# Designing against phase and property heterogeneities in additively manufactured titanium alloys

Jingqi Zhang [1,7], Yingang Liu [1,7], Gang Sha [2] ✉, Shenbao Jin[2], Ziyong Hou [3,4,5], Mohamad Bayat[6], Nan Yang [1], Qiyang Tan [1], Yu Yin [1], Shiyang Liu[1], Jesper Henri Hattel[6], Matthew Dargusch[1], Xiaoxu Huang [3,4] ✉ & Ming-Xing Zhang [1] ✉

Additive manufacturing (AM) creates digitally designed parts by successive addition of material. However, owing to intrinsic thermal cycling, metallic parts produced by AM almost inevitably suffer from spatially dependent heterogeneities in phases and mechanical properties, which may cause unpredictable service failures. Here, we demonstrate a synergistic alloy design approach to overcome this issue in titanium alloys manufactured by laser powder bed fusion. The key to our approach is in-situ alloying of Ti−6Al−4V (in weight per cent) with combined additions of pure titanium powders and iron oxide ($Fe_2O_3$) nanoparticles. This not only enables in-situ elimination of phase heterogeneity through diluting V concentration whilst introducing small amounts of Fe, but also compensates for the strength loss via oxygen solute strengthening. Our alloys achieve spatially uniform microstructures and mechanical properties which are superior to those of Ti−6Al−4V. This study may help to guide the design of other alloys, which not only overcomes the challenge inherent to the AM processes, but also takes advantage of the alloy design opportunities offered by AM.

Unlike conventional metal manufacturing processes such as casting and machining, additive manufacturing (AM) builds the digitally designed part up layer by layer via melting the feedstock (such as powder or wire) with a high energy source (for example, laser, electron beam or plasma arc)[1,2]. This unique feature of AM processes is a double-edged sword. On the one hand, it offers the possibility of producing desirable shapes, microstructures, and properties that cannot otherwise be achieved using conventional manufacturing methods[3–8]. On the other hand, the intrinsic steep thermal gradient, the high cooling rate in conjunction with the complex thermal history

typically encountered during AM often results in porosity, elemental segregation, columnar grains, and heterogeneously-distributed phases in the microstructure[9–12] – either in solidification or through subsequent solid-state phase transformations – which lead to non-uniform mechanical properties at different locations of the built metal part[13–17]. The issues related to the porosity, elemental segregation, and columnar grains have been effectively addressed through manipulation of processing parameters and/or alloy compositions[18–20]. However, as the phase inhomogeneity almost inevitably occurs in the alloys that undergo solid-state phase transformations after solidification during

[1]School of Mechanical and Mining Engineering, The University of Queensland, St. Lucia, Brisbane, Australia. [2]Herbert Gleiter Institute of Nanoscience, School of Materials Science and Engineering, Nanjing University of Science a nd Technology, Nanjing, China. [3]International Joint Laboratory for Light Alloys (Ministry of Education), College of Materials Science and Engineering, Chongqing University, Chongqing, China. [4]Shenyang National Laboratory for Materials Science, Chongqing University, Chongqing, China. [5]Department of Materials Science and Engineering, KTH-Royal Institute of Technology, Stockholm, Sweden. [6]Department of Mechanical Engineering, Technical University of Denmark, Lyngby, Denmark. [7]These authors contributed equally: Jingqi Zhang, Yingang Liu. ✉e-mail: gang.sha@njust.edu.cn; xiaoxuhuang@cqu.edu.cn; mingxing.zhang@uq.edu.au

AM, it remains a long-standing challenge to achieve uniform mechanical properties. Such phenomena are more pronounced in additively manufactured metallic components with complex geometries[21], which incorporate regions that respond differently to mechanical loading, thereby causing unpredictable service failures.

Ti−6Al−4V is one of the typical alloys which exhibit spatial variation of phases along the building direction while being additively manufactured[22–25]. During the AM process, such as laser powder bed fusion (L-PBF) (Fig. 1a), after the first layer is solidified, Ti−6Al−4V undergoes solid-state β (body-centred cubic structure) → α′ (hexagonal closed-packed structure) martensitic transformation due to the high cooling rate. As the successive layers are added, the acicular α′ martensite that was initially formed decomposes to lamellar (α + β) microstructures under extensive thermal cycles (Fig. 1a). Therefore, the microstructure of Ti−6Al−4V fabricated by L-PBF is commonly reported to feature spatially dependent phases along the building direction, with acicular α′ martensite on the top surface whereas partially or fully stabilized lamellar (α + β) microstructures forming in the lower regions[23–25]. Such a graded phase distribution is also confirmed by scanning electron microscope (SEM) (Fig. 1b and Supplementary Fig. 1a, b) and X-ray Diffraction (XRD) (Supplementary Fig. 2) in this work (Methods). To reveal the influence of the phase inhomogeneity on mechanical properties, we performed tensile testing of the L-PBF produced Ti−6Al−4V specimens along both vertical and horizontal

directions at room temperature (Methods). The as-fabricated Ti−6Al−4V exhibits similar strength but highly scattered ductility along both directions (Fig. 1c). In particular, the tensile ductility (in terms of the tensile elongation to failure) along the horizontal direction varies markedly from 9.4% to 17.6%, with the lowest value observed on the top surface. This trend, coupled with the detailed microstructural analysis (Supplementary Figs. 3–5 and Supplementary Note 1), reveals that the spatial phase distribution is the most likely cause of highly scattered ductility observed here. This observation is also consistent with the common belief that acicular α′ martensite generally results in inferior ductility compared with the lamellar (α + β) microstructure because of its inability to resist crack initiation[24,26]. Over the past decade, a wealth of studies have been carried out to eliminate the undesired α′ martensite in additively manufactured Ti−6Al−4V by L-PBF, which are based on the strategy of either process control or alloy design. The former strategy typically involves the manipulation of thermal cycling of L-PBF to trigger the intrinsic heat treatment (IHT)[27], which promotes in-situ martensite decomposition[24,27]. However, due to the limited or absent thermal cycles that the top layers undergo, acicular α′ martensite can only partially decompose or even remain[24,25]. Therefore, the phase inhomogeneity along the building direction cannot be eliminated. Although post-AM heat treatment is often performed to homogenize the microstructure[28], it unfortunately, lengthens the production cycle

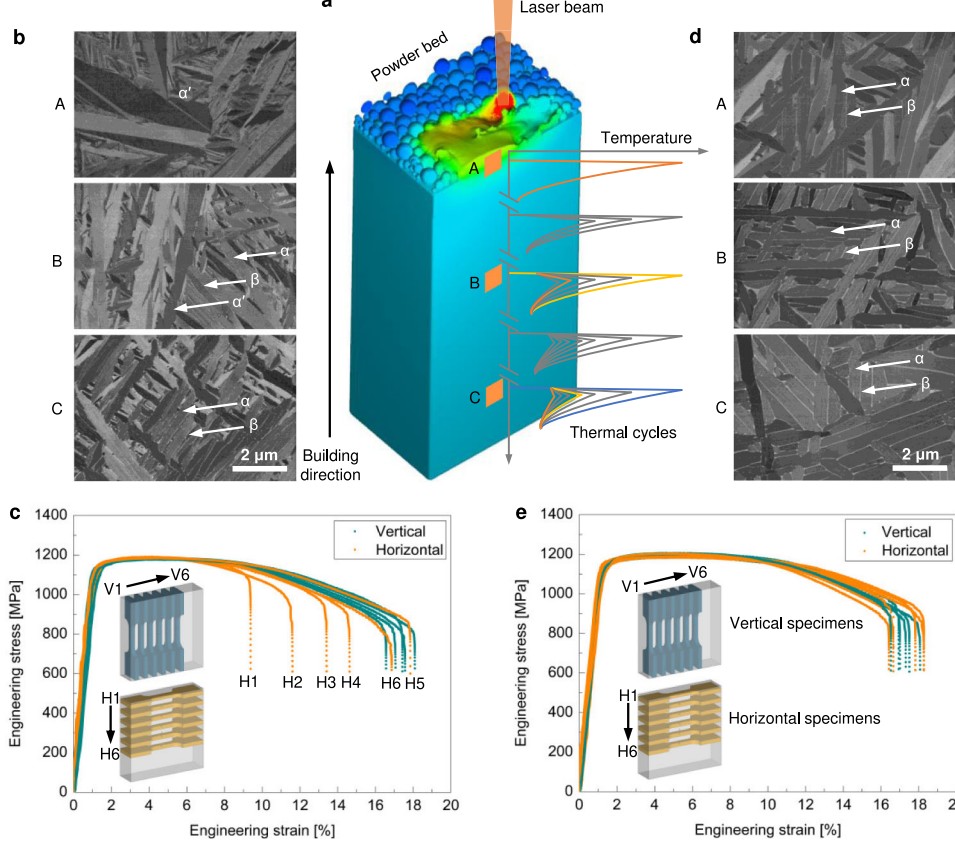

**Fig. 1 | Comparison of microstructures and tensile properties of Ti−6Al−4V and the newly developed alloy (25Ti−0.25O) fabricated via laser power bed fusion (L-PBF). a** Schematic of the L-PBF process and the intrinsic thermal cycles that different locations of the fabricated part undergo. **b** Scanning electron microscopy (SEM)-backscattered electrons (BSE) micrographs showing spatially dependent phases in Ti−6Al−4V along the building direction (BD) (lower magnifications in Supplementary Fig. 1). Note that the top surface is predominantly composed of acicular α′ martensite. The lower region shows a partial decomposition of α′ martensite due to more thermal cycles. It can be seen that α′ martensite, α phase, and thin β film are presented, as marked with white arrows. The bottom region exhibits

a well-defined lamellar (α + β) microstructure. **c** Tensile engineering stress-strain curves of Ti−6Al−4V along the vertical and horizontal directions. Inset, schematic of preparation of vertical and horizontal tensile specimens from the as-built parts. The horizontal tensile specimens are marked from H1 to H6 along the building direction. **d** SEM-BSE micrographs showing homogeneous lamellar (α + β) microstructure in the newly developed 25Ti−0.25O alloy (lower magnifications in Supplementary Fig. 1). The well-defined lamellar (α + β) microstructure can be observed from the top surface to the bottom region. **e,** Tensile engineering stress-strain curves of 25Ti−0.25O alloy along the vertical and horizontal directions. Inset, the preparation of the tensile specimens is the same as that of Ti−6Al−4V.

and influences the effectiveness of AM processes[29]. Hence, it is highly desirable to eliminate the phase inhomogeneity in the first place. Alternatively, in-situ alloying of Ti−6Al−4V with β stabilizing elements – for example, Mo[30] – through elemental powders allows for the formation of a full β phase, which leads to high ductility (albeit at the expense of strength loss). However, the resulting un-melted additive particles or dramatic elemental segregation may raise the concern of achieving non-uniform and unreproducible mechanical properties[31].

Here, we demonstrate a synergistic alloy design approach that enables in-situ elimination of the phase inhomogeneity in titanium alloys produced by L-PBF through combined additions of commercially pure titanium (CP−Ti) powders and $Fe_2O_3$ nanoparticles to Ti−6Al−4V feedstock. In sharp contrast to Ti−6Al−4V (Fig. 1b) showing significant phase variations along the building direction, the newly designed alloy – for example, the one with 25 wt % CP−Ti and 0.25 wt % $Fe_2O_3$ additions (hereafter referred to as 25Ti−0.25O and the other newly developed alloys are denoted in the same way) at a comparable strength level to Ti−6Al−4V – exhibits homogeneous lamellar (α + β) microstructures across the as-fabricated part (Fig. 1d and Supplementary Fig. 1c, d). This homogeneous microstructure results in uniform tensile properties along both vertical and horizontal directions (Fig. 1e). We further show that our alloy design approach is applicable to geometrically complex components in which the homogeneous lamellar (α + β) microstructures can also be achieved.

## Results

### Alloy design

The key to our design principle is to decrease the content of V with a low tracer diffusivity in Ti−6Al−4V whilst introducing small amounts of Fe with much higher tracer diffusivity, which enables faster elemental partitioning and thereby encourages in-situ formation of the lamellar (α + β) microstructures during L-PBF. This distinguishes our work from existing studies that promote grain refinement of prior-β grains in solidification through high Fe additions (Supplementary Note 2). It has been recognized that the martensite decomposition in Ti−6Al−4V involves the diffusive partitioning of α-stabilizing and β-stabilizing elements[27,29]. Specifically, Al (the α-stabilizer) accumulates in the α or α' phase, whereas V (the β-stabilizer) gets rejected from α' martensite and diffuses at the lattice defects. The kinetics of martensite decomposition strongly depend upon the elemental diffusivity. If we could reduce the V concentration whilst introducing a much stronger partitioning element – for example, Fe, Ni, or Co – the diffusivities of which are almost two orders of magnitude higher than V[32,33], then it is reasonable to expect that significant elemental partitioning can take place during the cooling process, thereby leading to the desired lamellar (α + β) microstructures rather than α' martensite that is formed by diffusionless transformation. To this end, we use a binary additive approach, that is, CP−Ti powders as the main additive and $Fe_2O_3$ particles as the trace additive. We introduce CP−Ti to Ti−6Al−4V to dilute the concentration of V. Moreover, as demonstrated by the thermo-dynamic calculation (Supplementary Fig. 6), an additional benefit of CP−Ti addition is that it decreases the Al concentration, thus suppressing the tendency for the formation of brittle $α_2$-$Ti_3Al$ phase that can be triggered by IHT in Ti−6Al−4V fabricated by L-PBF[25]. We select $Fe_2O_3$ particles as the trace additive for two reasons. First, Fe exhibits much higher diffusivity than V in the β phase[32,33]. Second, the dramatic solid-solution strengthening of O in titanium enables the offset of strength loss due to the dilution of Ti−6Al−4V through CP−Ti addition (Supplementary Note 3). It is worth noting that marrying the fast diffusion elements (for example, Fe, Ni, and Co) with the strong interstitial strengthening solutes (that is, N, O, and C) in titanium offers a diverse choice of trace additive. Here in this work, we choose $Fe_2O_3$ as an example of our alloy design practice because it is cost-effective, and

its red color can serve as an indicator for the feedstock preparation process, as will be described below.

### Feedstock preparation

We then prepared the powder feedstock for L-PBF. Conventional mechanical mixing has been commonly used in feedstock preparation but often suffers from blending inhomogeneity due to the agglomeration of additive particles. In this work, the homogeneous distribution of $Fe_2O_3$ particles in the titanium feedstock is critical for achieving uniform mechanical properties because of the strong oxygen sensitivity in titanium[34]. Here, we used a surface engineering approach to synthesize the $Fe_2O_3$-doped titanium feedstock (Fig. 2a). This approach is based on the layer-by-layer (LbL) assembly technology that has been used to make multilayer thin films[35]. However, here we did not use the functional multilayer deposition but adopted the alternating adsorption process to induce stable charged surfaces of both Ti−6Al−4V and CP−Ti powders, which facilitated the adhesion of $Fe_2O_3$ nanoparticles. More importantly, we utilized zeta potential measurement to determine the surface charges of titanium powers and $Fe_2O_3$ nanoparticles, which is critical for the adsorption sequence (Methods and Fig. 2a). In contrast to the mechanically-mixed powders, which exhibit the dramatic agglomeration of $Fe_2O_3$ nanoparticles even at the macroscale (Fig. 2d), the feedstock prepared by the surface engineering approach shows a fairly uniform distribution of $Fe_2O_3$ nanoparticles on the surface of individual titanium powder, as evidenced by SEM together with energy-dispersive X-ray spectroscopy (EDS) mapping (Fig. 2e, f).

### Mechanical properties

We produced a series of titanium alloys by tuning CP−Ti and/or $Fe_2O_3$ addition levels (Supplementary Table 1) and carried out tensile testing of the newly developed alloys under exactly the same conditions as Ti−6Al−4V. In contrast to Ti−6Al−4V in this work (Fig. 1c), a striking mechanical response of the newly developed alloys is the exceptional uniform ductility in both vertical and horizontal directions (Fig. 3a, b). We notice that the change of strength-ductility combinations does not come with the loss of uniform mechanical response. Unlike the conventional wisdom which typically uses a single additive, the binary additive approach that we adopted here offers more freedom to tailor mechanical properties over a wide range – a yield strength from 831.4 ± 2.7 MPa to 1,220.8 ± 6.5 MPa and an elongation to failure from 26.7 ± 0.6% to 13.7 ± 0.9% – by simply tuning the addition level of each additive. For example, we can achieve very high strength – which is comparable to, or even higher than, those reported for Ti−6Al−4V fabricated by L-PBF and L-PBF plus heat treatment (L-PBF + HT) – yet still with higher ductility (for example, 25Ti−0.50O and 25Ti−0.25O alloys in Fig. 3c). Besides, by either decreasing $Fe_2O_3$ or increasing CP−Ti addition level (that is, 50Ti−0.50O, 50Ti−0.25O and 75Ti−0.25O alloys in Fig. 3c), we can obtain excellent ductility higher than 20%, which is double the minimum requirement of Ti−6Al−4V recommended for critical structural applications (that is, 10%)[24]. Overall, the tensile properties of our alloys not only profoundly outperform those of the conventionally manufactured Ti−6Al−4V (for example, mill annealed, and solution-treated and aged)[36], but also significantly extend the current strength-ductility limit reported for Ti−6Al−4V by L-PBF (without[24,29,37–39] and with additional heat treatment[37–41]), Ti−6Al−4V based composite by L-PBF[30], electron beam-based powder bed fusion (EB-PBF)[42] (Supplementary Note 4) and directed energy deposition (DED)[43]. It should be noted that the main aim of this study is to demonstrate the feasibility of our design strategy to develop alloys matched to the AM processes. Hence, our experimental work on the selected addition levels of CP−Ti (25 wt %, 50 wt %, and 75 wt %) and $Fe_2O_3$ (0.25 wt % and 0.50 wt %) is simply an example of our alloy design practice. A wide variety of other alloy compositions and

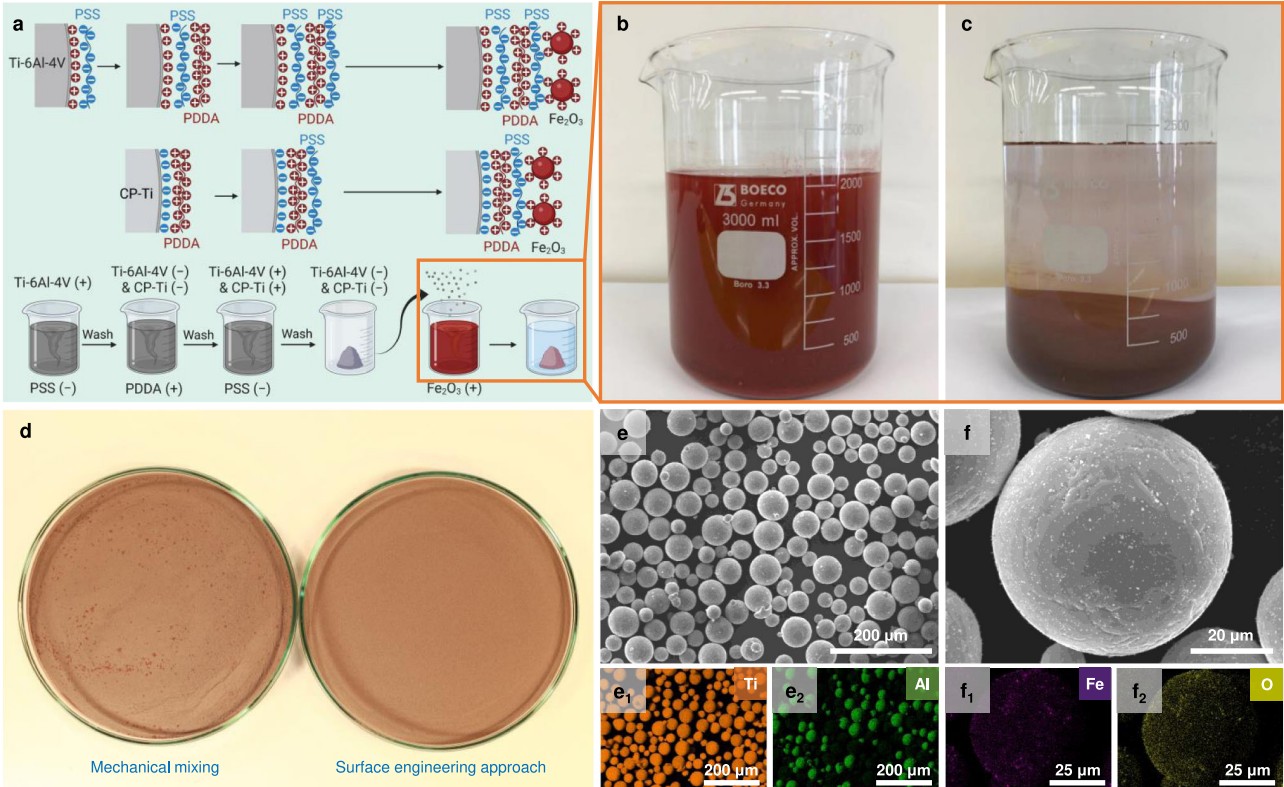

**Fig. 2 | Feedstock preparation and characterization. a** Schematic of the feedstock preparation process via the surface engineering approach (Methods). Here, PSS and PDDA are poly(sodium 4-styrene sulfonate) and poly (diallyldimethylammonium chloride), respectively. The schematic diagram was created with BioRender.com. **b**, **c** The $Fe_2O_3$ suspension before (**b**) and after (**c**) the addition of processed Ti−6Al−4V and CP−Ti powders as shown in (**a**). **d** Comparison of feedstocks prepared by mechanical mixing and the surface engineering approach. **e** and **f** SEM and energy-dispersive X-ray spectroscopy (EDS) images showing the homogeneous distributions of both CP−Ti (**e**) and $Fe_2O_3$ (**f**) in the powder feedstock for the designed 50Ti−0.25O alloy.

mechanical properties can be achieved through tuning the CP−Ti and/ or $Fe_2O_3$ addition levels.

## Elemental and microstructural characterizations

To better understand the exceptional uniform mechanical properties, we carried out detailed elemental and microstructural characterizations. The newly developed alloys show highly homogeneous elemental distributions of Al and V without any macro-segregation (Supplementary Fig. 7). Moreover, like previous microstructural analysis of Ti−6Al−4V, our electron backscatter diffraction (EBSD) (Supplementary Fig. 8), microfocus computed tomography (Micro-CT) (Supplementary Fig. 9) and SEM (Supplementary Fig. 10) characterizations rule out the possibility of porosity and columnar grain as the potential sources of mechanical inhomogeneity in the newly developed alloys. In contrast to Ti−6Al−4V, our newly developed alloys show uniform lamellar (α + β) microstructures from the bottom region to the top surface (Fig. 1d and Supplementary Fig. 11). Such a microstructural uniformity eliminates the most plausible origin of mechanical inhomogeneity as observed in Ti−6Al−4V in this work and other studies (Supplementary Note 4).

To gain a deeper insight into the formation of lamellar (α + β) microstructures on the top surface of the newly developed alloys, we performed the kinetics simulation of martensite decomposition using the DICTRA (DIffusion-Controlled TRAnsformation) software (Methods). The simulation is based on the experimental L-PBF process, in which the top fused layer that does not experience any laser re-heating cools down after martensitic transformation (Supplementary Fig. 12a, b). In the case of Ti−6Al−4V, α′ martensite was retained (Fig. 1b) because martensite decomposition requires sufficient thermal cycling and time. In contrast, the newly developed alloy exhibits

significant diffusion partitioning of Fe upon cooling. It is evident that Fe shows a much stronger partitioning tendency in the β phase than V (Supplementary Fig. 12c−e) due to its significantly high diffusivity (Supplementary Fig. 13). Such a fast diffusion partitioning of Fe, along with the strong β-stabilizing effect, is essential to form the β phase, thereby leading to the lamellar (α + β) microstructures. On the other hand, it is found that the α stabilizers Al and O are accumulating in the α′ phase as the cooling process proceeds (Supplementary Fig. 14a, b). The dynamic element partitioning of Fe, V, Al, and O is in line with the conclusion drawn from the APT characterization by Haubrich et al.[27].

We observed the lamellar (α + β) microstructure in the selected newly developed alloy (50Ti−0.50O) using transmission electron microscopy (TEM) (Supplementary Fig. 15a, b). The STEM-EDS images clearly reveal that the β phase is enriched in both Fe and V while it is depleted in Al (Supplementary Fig. 15c). We also performed compositional analysis using atom probe tomography (APT) (Fig. 4 and Supplementary Video 1). It is observed that increasing $Fe_2O_3$ doping level leads to the transition of discontinuous β film to a continuous and relatively thick β phase (Fig. 4a, b). The element partitioning ratio $K^{-1}$, which was derived from APT proximity histograms (Fig. 4c, d), shows that Fe has much higher partitioning ratios than V (Fig. 4e), indicating a much stronger partitioning tendency of Fe in the β phase. This is in line with the expectation from our DICTRA simulation (Supplementary Fig. 12c−e) and other experimental work on Ti−6Al−4V[27]. In addition, the concentrations of Fe in the β phase are in the range of 6.5 − 8.6 at %. These values are close to those reported for post-AM heat-treated Ti−6Al−4V[27], indicating that Fe might almost reach its equilibrium state in the β phase. The elemental analysis further supports that the faster partitioning ratio of Fe allows for in-situ formation of lamellar (α + β) microstructures during fabrication.

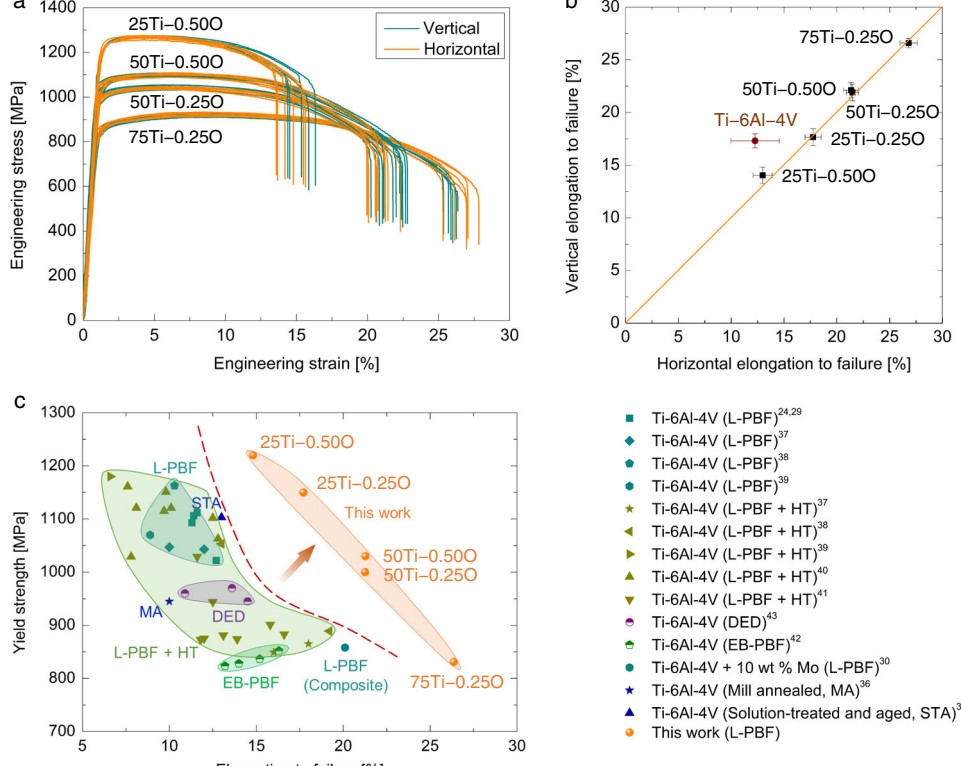

**Fig. 3 | Mechanical properties of the newly developed alloys fabricated by L-PBF. a** Tensile engineering stress-strain curves along the vertical and horizontal directions, indicating uniform tensile properties. **b** The elongation to failure of Ti−6Al−4V and the newly developed alloys along the vertical and horizontal directions. The significant data deviation of Ti−6Al−4V from the orange line indicates a high degree of scattering in ductility. The error bars represent the standard deviation of the mean. **c** Comparison of the tensile properties of our alloys with those of Ti−6Al−4V (and Ti−6Al−4V based composite) fabricated by L-PBF (both in as-built and heat-treated states), directed energy deposition (DED) and electron beam-based powder bed fusion (EB-PBF).

Promoting in-situ α′ martensite decomposition through process control does offer a pathway to high-performance Ti−6Al−4V[24]. However, this strategy may not be applicable to components with small size and/or complex geometry because the thermal history that the material experiences is strongly size and geometry-dependent. To demonstrate the applicability of our approach, we further fabricated geometrically complex components with different sizes (Supplementary Fig. 16a, b). The microstructural examination validates homogeneous lamellar (α + β) microstructures rather than acicular α′ martensite even in the downscaled part (Supplementary Fig. 16c, d), which is in contrast to that made from Ti−6Al−4V by L-PBF[21]. This observation confirms that our approach enables the fabrication of geometrically complex components yet with uniform (α + β) microstructures.

In this work, we have designed and additively manufactured a series of titanium alloys that possess exceptional tensile properties without notable mechanical inhomogeneity. We have shown that the typical and undesired phase inhomogeneity in titanium alloys – which is associated with thermal cycling inherent to AM – can be eliminated through rational alloy design. The key to our approach lies in the partitioning of alloying elements in phase decomposition, which is a common feature of solid-state phase transformations in metallic materials[44,45]. We expect that the newly developed titanium alloys could be candidate materials in applications where titanium alloys with uniform mechanical properties are demanded. This requires a comprehensive evaluation of other mechanical properties (such as fatigue properties and creep resistance) and corrosion resistance (Supplementary Note 2). Furthermore, unlike previous studies, which have mainly focused on grain refinement (through alloy design) and/or defect control (via processing optimization), our work demonstrates that addressing the phase heterogeneity is of equal, if not greater,

importance to achieve the desired uniform mechanical properties. Since the phase heterogeneity due to the solid-state thermal cycling has been reported in a wide variety of metallic materials fabricated by different AM technologies[12,46–49], we believe that our design strategy may help the development of other metallic alloys specifically for AM with uniform mechanical properties.

## Methods
### Feedstock preparation
Ti−6Al−4V ELI (Grade 23, SLM Solutions Group AG, Germany) and CP−Ti (Grade 1, Advanced Powders and Coatings, Canada) powders used in this work have a spherical shape both with a particle size range of 20−63 μm. The iron (III) oxide ($Fe_2O_3$) (Sigma-Aldrich, Germany) particles have a particle size of less than 5 μm. The unit price of $Fe_2O_3$ particles is much lower than those of Ti−6Al−4V and CP−Ti powders.

To prepare the feedstock for L-PBF, surface charges of Ti−6Al−4V, CP−Ti, and $Fe_2O_3$ were first measured in deionized water by using a Malvern Zetasizer (ZS90, Malvern Instruments, UK). It is found that Ti−6Al−4V powders yield a high positive zeta potential of 53.50 ± 0.54 mV while the measured zeta potential of $Fe_2O_3$ particles is relatively low, with a value of 9.12 ± 0.18 mV. Conversely, CP−Ti powders are negatively charged with a zeta potential of −17.97 ± 1.94 mV. The adsorption sequence is based on the zeta potential measurement. Because Ti−6Al−4V and CP−Ti powders are oppositely charged (Fig. 2a), the positively charged Ti−6Al−4V powders were first mixed in the solution containing 8 mg mL$^{-1}$ poly(sodium 4-styrene sulfonate) (PSS, average molecular weight: Mw = ∼ 70,000, powder, Aldrich) which shows negative charges. A monolayer of the polyanion was adsorbed, and thus the surface charges of Ti−6Al−4V powders were reversed. After being rinsed in deionized water (the aim of rinsing was

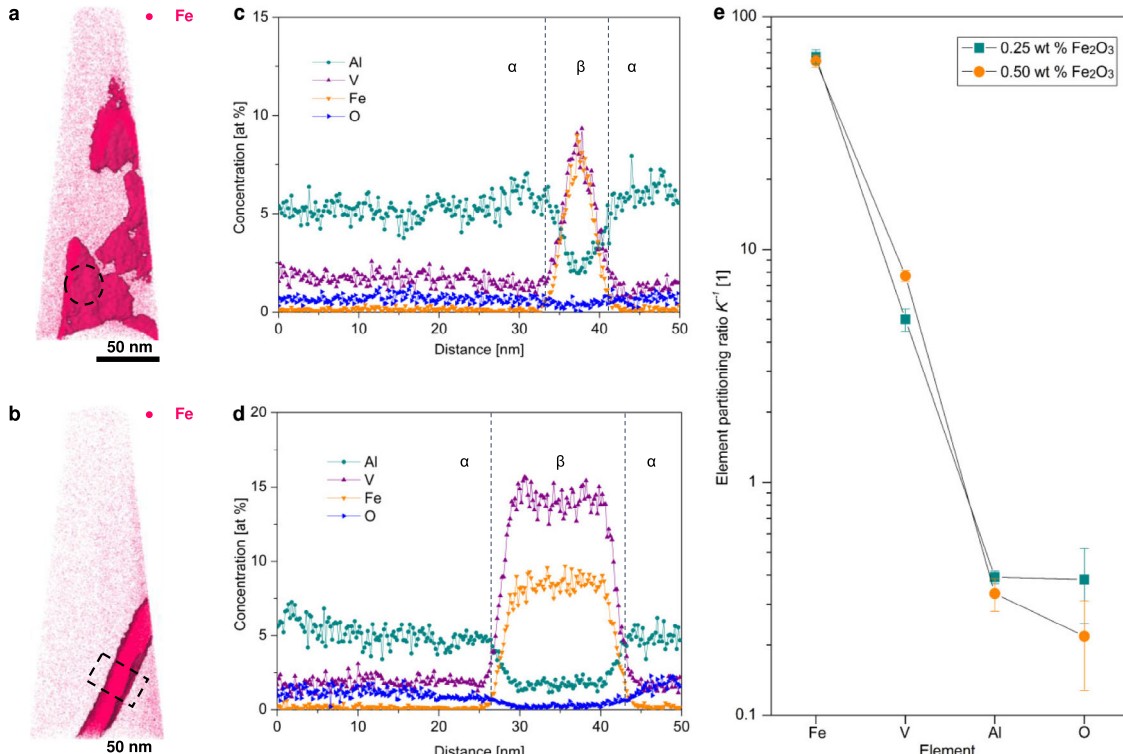

**Fig. 4 | Atom probe tomography (APT) characterization of the newly developed alloys. a**, **b** 3D reconstruction (also in Supplementary Movie 1) of Fe distribution in the samples with different $Fe_2O_3$ addition levels (A, 50Ti−0.25O, B, 50Ti−0.5OO). **c**, **d** Proximity histograms across the β phase in (**a**) (marked with dashed black cycle) and (**b**) (marked with black dashed rectangle) showing enrichment (Fe and V) and depletion (O and Al) of alloying elements in the β phase. **e** The element partitioning ratio $K^{-1}$ (which is defined by $C_\beta/C_\alpha$, where $C_\beta$ is the concentration inside the enriched features and $C_\alpha$ outside these regions) of Fe, V, Al, and O derived from the APT data. Note that high partitioning ratios suggest a strong accumulation of an element in these regions, while $K^{-1} < 1$ indicates elemental depletion. It can be seen that Fe shows much higher partitioning rates than V. The error bars denote the standard deviation of the mean.

to remove the loosely adsorbed PSS from the powders), Ti−6Al−4V powders together with CP−Ti powders (both show a negative charge) were immersed in the positively charged 8 mg mL$^{-1}$ poly(-diallyldimethylammonium chloride) (PDDA, average molecular weight: Mw = 200,000−350,000, 20 wt % in $H_2O$, Aldrich) solution. Again, the adsorption of the PDDA monolayer led to the reversal of the surface charge. This process was repeated so that both Ti−6Al−4V and CP−Ti powders were negatively charged due to the adsorption of the PSS monolayer (Fig. 2a). Finally, the rinsed Ti−6Al−4V and CP−Ti powders were mixed in the red-colored $Fe_2O_3$ suspension (the liquid used to suspend $Fe_2O_3$ particles is deionized water). After stirring for 30 min, sedimentation took place, that is, the powers that showed the red color settled down at the bottom of the beaker and left a clear layer of liquid above (Fig. 2c), indicating successful dispersion of $Fe_2O_3$ particles in the titanium powders. After removal of the liquid, the powders were dried in a vacuum drying oven (Labec, Australia) for more than 10 h. Following the feedstock preparation, the powder mixtures were characterized by scanning emission microscopy (SEM, JEOL JSM-6610, Japan) equipped with energy dispersive X-ray spectroscopy (EDS).

For the purpose of comparison, powders mixtures were also prepared by mechanical mixing. Ti−6Al−4V and CP−Ti powders with 0.5 wt % $Fe_2O_3$ particles were mixed using a Tubular shaker mixer (Willy A. Bachofen AG, Switzerland) for 60 min.

### Additive manufacturing
Laser powder bed fusion was performed on an SLM®125HL machine (SLM Solutions Group AG, Germany) equipped with a 1,060 nm wavelength IPG fibre laser (max laser power of 400 W and a laser spot size of 80 μm). Prior to L-PBF, the titanium substrate plate was preheated up to 200 °C under a high-purity argon atmosphere (99.997%). L-PBF was carried out when the oxygen level was reduced below 0.02 vol%. Parameter optimization was performed by using Ti−6Al−4V powders, with the aim of achieving a very high density and interrupting the columnar grains. The "meander" scanning strategy with an initial 45° scanning angle and an 67° rotation between each layer was adopted. The building time of each layer (exposure time and coating time) was kept constant at 15 s. A schematic illustration of the scanning strategy is provided in Supplementary Fig. 17. The refined processing parameters were 350 W laser power, 1,400 mm s$^{-1}$ scanning speed, 30 μm layer thickness, and 120 μm hatch spacing.

For both Ti−6Al−4V and the newly developed alloys, titanium parts with the dimension of 40 mm (length) × 10 mm (width) × 40 mm (height) were built with a 2 mm supporting structure on a 50 mm × 50 mm titanium substrate plate. For each composition, two titanium parts were simultaneously built on the substrate plate so that the thermal histories of these parts encountered during L-PBF were essentially the same. The chemical compositions of the newly developed alloys were measured by using inductively coupled plasma atomic emission spectroscopy (ICP-AES) for metallic elements and by using LECO combustion analysis for non-metallic elements, as listed in Supplementary Table 1.

### Mechanical testing
For tensile testing, dogbone-shaped tensile specimens with gauge dimensions of 10 mm (length) × 2.5 mm (width) × 2 mm (thickness) were machined from the as-built parts along both the vertical and horizontal directions by electrical discharge machining (Inset in Fig. 1c). The geometry of the tensile specimen was adopted from ref. 50. The tensile specimens were carefully marked to keep track of

their location on the as-built parts. Prior to the tensile testing, tensile specimens were mechanically polished down to 4,000 grit size to eliminate the post-machining surface roughness. Room-temperature tensile tests were carried out on an electromechanical universal testing machine (Model 5584, Instron Inc., USA) equipped with a 10 kN load cell at a constant strain rate of 0.001 s⁻¹. The strain evolution of tensile specimens was tracked using an Instron AVE2 non-contacting video extensometer (Instron Inc., USA) with a data rate of 490 Hz and a resolution of 0.5 μm in the axial dimension. Six specimens were tested for each group. Following the tensile testing, the fracture surface was analysed by using a SEM (JEOL JSM-6610, Japan).

## Scanning electron microscopy

For phase analysis, samples were cut from different locations along the building direction of the as-fabricated parts and were mechanically polished using Struers OP-S suspension containing 20 vol % of $H_2O_2$ for 30 min without any chemical etching. The microstructure was characterized in backscattered electron (BSE) mode using an FEI Scios Dual Beam system (Thermo Fisher Scientific Inc., USA) equipped with a concentric backscattered (CBS) detector under 3 kV accelerating voltage, 1.6 nA probe current, and 5.5 mm working distance. For SEM-EDS mapping of as-fabricated parts, the samples prepared for phase analysis were then characterized by using an SEM (JEOL JSM-6610, Japan).

## X-ray diffraction

X-ray diffraction (XRD) analysis was conducted on a D8 ADVANCE X-ray diffractometer (Bruker, Germany) (Cu radiation source) operated at 40 kV and 40 mA with a step size of 0.02°.

## Microfocus computed tomography

Microfocus computed tomography (Micro-CT) was performed on the tensile specimens after tensile testing using a Micro-CT system (diondo d2, Germany), with a spatial resolution of 4 μm. Micro-CT characterization was carried out on the grip and gauge regions of the specimens, which allowed for direct observation of the porosity distribution in both the as-built and post-testing states. Selected parts of the horizontal specimen were also characterized at a spatial resolution of 2 μm.

## Electron backscatter diffraction

Samples for EBSD characterization were ground and mechanically polished using Struers OP-S suspension containing 20 vol % $H_2O_2$ for 30 min. Final electropolishing was carried out at room temperature using a voltage of 20 V for 240 s in Struers electrolyte A3. EBSD characterization was performed on a SEM (JEOL JSM-7800F, Japan) with a step size of 0.3 μm. The EBSD data was analysed using OIM Analysis 7.3 software. The prior-β grain structure was reconstructed using the ARPGE software package[51].

## Transmission electron microscopy

Samples for transmission electron microscope (TEM) observations were prepared using a FEI Scios Dual Beam system (Thermo Fisher Scientific Inc., USA). TEM characterization was performed on a FEI Tecnai G2 F20 TEM (Thermo Fisher Scientific Inc., USA) operated at an acceleration voltage of 200 kV in both TEM and scanning TEM (STEM) modes.

## Atom probe tomography

The nanoscale elemental distributions in the fabricated parts were analysed by atom probe tomography (APT) using a local electrode atom probe CAMECA LEAP 4000X SI. Samples for APT characterization were prepared using a Zeiss Auriga dual-beam focused ion beam (FIB, Carl Zeiss Microscopy, Germany), with standard FIB lift-out procedures[52]. The data was acquired under a high vacuum of $2 \times 10^{-11}$ torr, at a specimen temperature of 40 K, a pulse repetition rate of 200 kHz, and UV laser energy of 40 pJ. The APT data were reconstructed using a CAMECA integrated visualization and analysis software (IVAS 3.8.2) (CAMECA Scientific Instruments, USA), with a tip profile method in reference to the SEM image of each tip.

## Electrochemical measurements

To evaluate the corrosion resistance, electrochemical measurements of the selected samples made from 25Ti−0.25O and 25Ti−0.50O were carried out at room temperature in 3.5 wt % NaCl solution. Ti−6Al−4V was also tested as the reference material. All electrochemical measurements were performed on a potentiostat (PARSTAT® 2273 Princeton, Applied Research, USA) with a conventional three-electrode cell incorporating a platinum sheet counter electrode and a saturated calomel reference electrode. The sample surfaces (with an area of 1.2 cm²) were mechanically polished down to 4000 grit size before testing and served as the working electrode. The open circuit potential (OCP) of each sample was recorded for 2 h, followed by electrochemical impedance spectroscopy (EIS) and potentiodynamic polarisation (PDP) during the electrochemical measurement. EIS was carried out with the frequency range between $10^{-2}$ Hz and $10^{5}$ Hz using a 10 mV sinusoidal perturbating signal. The analysis of EIS data was performed using an ZView software. The PDP measurement was carried out at the scan rate of 1 mV s⁻¹, in the potential range of −250 mV vs OCP, and completed at +5000 mV vs OCP. Three repeatable measurements were taken from each sample group.

## Simulation and calculation

For Ti−6Al−4V, the top surface of the L-PBF-produced part did not experience any laser reheating or very limited thermal cycles. Therefore, α′ martensite was retained, and the phase heterogeneity in Ti−6Al−4V was formed. According to our design strategy, it is expected that, in the newly developed alloys, a significant level of Fe partitioning would take place in the top surface during cooling and the fraction of the β phase increases through the partitioning of Fe across the α′/β interface, thereby leading to the development of lamellar (α + β) microstructures. To qualitatively evaluate the possibility of in-situ formation of lamellar (α + β) microstructures, a kinetics simulation was performed using the DICTRA software. Prior to the simulation, two key inputs should be determined that is the martensite start temperature $M_s$ of the newly designed alloys and the cooling profile that the top surface experienced below $M_s$.

**Determination of martensite start temperature $M_s$.** The martensite start temperature $M_s$ of the newly designed alloy was calculated based on the heterogeneous martensite nucleation theory developed by Olson and Cohen[53–55]. According to this theory, the critical condition for martensite to nucleate is given by:

$$\Delta G_{chem} + \Delta G_{mech} = -(W_f + G_0) \tag{1}$$

where $\Delta G_{chem}$ is the chemical driving force, $\Delta G_{mech}$ is the mechanical driving force, and the term ($\Delta G_{chem} + \Delta G_{mech}$) denotes the total driving force. The right-hand side term represents the critical driving force for martensite nucleation ($\Delta G_{crit}$) which includes the composition-dependent frictional work term ($W_f$) and a nucleation potency term ($G_0$).

For martensite formed by quenching, the driving force for martensitic transformation is only from the chemical contribution[56,57], and hence Eq. (1) becomes:

$$\Delta G_{chem} = \Delta G_{crit} = -(W_f + G_0) \tag{2}$$

In Eq. (2), the chemical driving force ($\Delta G_{chem}$) is given by:

$$\Delta G_{chem} = G_m(\alpha) - G_m(\beta) \tag{3}$$

where $G_m(\alpha)$ and $G_m(\beta)$ are the molar Gibbs energy of α and β phases, respectively. $\Delta G_{\text{chem}}$ was determined by using the TiGen (Titanium Genome) database[58]. The critical driving force for martensite nucleation is expressed by:

$$\Delta G_{\text{crit}} = -G_0 - \sum_i K_i x_i \qquad (4)$$

where $K_i$ is the athermal strength of solute $i$ and $x_i$ is the atomic fraction of solute $i$. The temperature at which the critical driving force equals the chemical driving force corresponds to $M_s$ temperature:

$$\Delta G_{\text{chem}} = \Delta G_{\text{crit}} \qquad (5)$$

It should be noted that the TiGen database does not include oxygen. Since Fe lowers $M_s$ temperature[59] while O raises it[60], the alloy compositions without $Fe_2O_3$ addition could be used for a rough estimate of $M_s$ temperatures. The calculated $M_s$ values are listed in Supplementary Table 2. These values are higher than the generally accepted $M_s$ temperature for Ti−6Al−4V (~800 °C[61]) and lower than that of CP−Ti (860 °C for Grade 2 CP−Ti[62]), indicating a reasonable estimation of $M_s$ temperature in this work.

**Multi-physics simulation.** The cooling pathway that the top surface underwent following fabrication cannot be readily measured. Hence, a single-track multi-physics simulation of L-PBF (Supplementary Fig. 11a) was performed to obtain the cooling curve for subsequent DICTRA calculation. This simulation was based on the finite volume method (FVM) framework in the commercial software Flow-3D[63]. The thermo-physical properties of 50Ti−0.50O alloy used in the simulation were calculated using Thermo-Calc software with the TCTI3 database, as listed in Supplementary Table 3. The L-PBF processing parameters for simulation were 350 W laser power, 1,400 mm s$^{-1}$ scanning speed, and 30 μm layer thickness, which is the same as those used in the experimental work.

**DICTRA simulation.** DICTRA simulation was performed using Thermo-Calc software implemented TCTI3 and MOBTI4 database. The initial simulation cell consists of an (α′+β) microstructure where the thickness of α′ and β is 1,000 nm and 10 nm, respectively. The constituent phases and their sizes for simulation are based on the previously reported experimental result[27]. The initial composition of the simulation cell was calculated based on the designed 50Ti−0.50O alloy using the chemical compositions of Ti−6Al−4V and CP−Ti powers as well as 0.50 wt % $Fe_2O_3$ addition level, which are provided in Supplementary Table 1. Given that the temperature of the titanium substrate plate was kept at 200 °C during fabrication and the temperature of the fabricated part could be raised above 500 °C due to energy input[24], the cooling profile of the top surface in the simulation was assumed to start from $M_s$ temperature to 500 °C and then maintained at 500 °C until 100 s (Supplementary Fig. 11b).

## Data availability
The data that support the findings of this study are available from the corresponding author on request.

## Code availability
All related codes are available from the corresponding authors on request.

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

## Acknowledgements

We thank W. Xu of Deakin University, J. Kruzic of the University of New South Wales, and M. Bermingham of The University of Queensland for fruitful discussions and valuable comments. We acknowledge J.Y. Yan of Thermo-Calc Software AB for assistance with thermodynamic calculation, H.X. Li of The University of Queensland for assistance in performing electrochemical measurements, and N. Mclean of The University of Queensland for careful proofreading. The authors from The University of Queensland gratefully acknowledge financial support from the Australian Research Council (ARC, grant number: DP210103162). The facilities and technical assistance of the Australian Microscopy & Microanalysis Research Facility at the Centre for Microscopy and Microanalysis (CMM), The University of Queensland, are also acknowledged. X.X. Huang and Z.Y. Hou acknowledge the financial support from the National Key Research and Development Program of China (2021YFB3702101) and the "111" Project (B16007) by the Ministry of Education and the State Administration of Foreign Experts Affairs of

China. The authors would like to acknowledge facility use and scientific and technical assistance from the Materials Characterization Facility at Nanjing University of Science and Technology.

## Author contributions

J.Z., M.-X.Z., and X.X.H. conceived the concept and designed the experiments. M.-X.Z., X.X.H., and M.D. supervised the project. J.Z. and Y.L. carried out the main experimental work. Z.H. performed thermodynamics a + -nd kinetics simulations. G.S. and S.J. conducted the APT characterization. M.B. and J.H.H. carried out the multi-physics simulation. N.Y. performed the electrochemical measurements. Q.T., Y.Y., and S.L. helped with the TEM and EBSD characterizations. J.Z., Y.L., M.D., G.S., J.H.H., X.X.H., and M.-X.Z. wrote and revised the manuscript. All authors contributed to the analysis and discussion of the data.

## Competing interests

The authors declare no competing interests.
