## [Peer Review File · Nature Communications]

Title: Designing against phase and property heterogeneities in additively manufactured titanium alloysREVIEWER COMMENTS

Reviewer #1 (Remarks to the Author):

In this manuscript, authors proposed a novel material designing strategy for additive manufacturing to achieve the phase and property uniformity in the AMed titanium alloy, using Ti-6Al-4V (Ti64) as an example. The critical and novel point in this strategy is to in-situ alloying of Ti64 with combined addition of pure titanium powders and iron oxide nanoparticle to tune the phase transformation kinetics during the additive manufacturing process. This strategy focusing on the uniformity of microstructure in the AMed titanium alloys is novel. In the past, the designing strategy of AMed titanium alloys mainly focused on the beta grain refinement (alloy design) or defect control (processing optimization). The current work on the other hand is focusing on the homogeneity of alpha+ beta microstructure in the AMed Ti-64 alloy. It is to avoid the alpha prime martensite phase formation during the AM process in the layers close to the top of the product, where limited thermal cycles occurred. Thus, the designing strategy is novel and the reported research is significantly beneficial to the understanding of processing-microstructure-property in the AMed Ti64 alloy.

However, there are several questions authors are required to answer clearly:

1. The newly developed alloy is not Ti-64, but a new Ti-Al-V-Fe-O alloy. So why it is an important alloy to study?

Ti-64 alloy occupies the largest amount in the titanium market, and thus, significant amount of effort is spent on the low-cost manufacturing of this important Ti-64 alloy. However, in this work, different solutes (some is large amount) have been added into Ti-64 and thus significantly change the alloy composition. In the reported work, 25, 50 and 75wt% of CP-Ti was added into Ti-64 with 0.25-0.50wt% Fe₂O₃. Thus, the alloy manufactured is not Ti-6-4 anymore, but a new Ti-Al-V-Fe-O alloy. So the question is even if the microstructure produced in the newly developed alloy is full of alpha + beta microstructure, why it is an important alloy or why people need to study and manufacture this alloy?

2. The characterization of alpha' and alpha microstructure in the Ti-6-4 and newly designed Ti-Al-V-Fe-O alloy needs further analysis.

Alpha' and alpha phases exhibit HCP structure with slightly different composition (and thus different lattice parameter). It seems in the manuscript, morphology difference characterized in the SEM BSE imaging is mainly used to distinguish the alpha' and alpha phases (in Fig. 1, supplemental Fig. 1, supplemental Fig. 10 and supplemental Fig. 12). So did authors use APT study the composition of the claimed alpha prime phase in the Ti-64? If so, did authors observe any composition change in the alpha prime phase in the AMed Ti-64? If there is any composition change, can it be claimed as alpha prime phase? The reasons behind these questions actually challenge the proposed designing strategy, there may be limited diffusion occurs between the claimed alpha prime phase and beta matrix. If so, the addition of Fe, fast diffuser, is not that critical in forming alpha phase, as claimed by the authors.

3. The claimed application of the approach to the beta titanium alloy needs further explanation.

Authors claim the proposed approach can be applied in the beta titanium alloys to trigger the omega phase and alpha phases in the building direction. However, Fe is a strong beta phase stabilizer and oxygen will impede the omega phase formation as well. Thus, I don't think the addition of Fe₂O₃ will promote the omega phase and alpha phase during the AM process in the building direction. So authors are required to introduce more details how the proposed strategy can be used in other titanium alloys.

4. The addition of Fe, the claimed key to the proposed approach, needs further discussion.

Recently, different phase transformation mechanisms have been proposed in the field of titanium alloys to explain the formation of alpha phase. Whether or not partitioning is required to form alpha phase is being challenged:

- 1) Physical Review B 74, 134114 (2006). The concept of "bainitic alpha" was proposed in this work and it was claimed that "the growth of bainitic alpha plates is partitionless".
- 2) Acta Materialia 60 (2012) 6247-6256. The concept of "pseudo-spinodal decomposition" was proposed that the structure and composition change in the formation of alpha may not occur simultaneously.

If diffusion is not required to form alpha microstructure in the titanium alloys, is it still necessary to add the fast diffuser (like Fe in the current work) or to manipulate the partitioning of alloying element in phase decomposition, which is the "key to our approach" claimed in the manuscript?

Reviewer #2 (Remarks to the Author):

This is an excellent contribution. The approach is novel, the methods and analysis is very well documented, the results on mechanical behavior quite interesting. My only recommendation for a minor modification is that the authors should point out also that while this approach is suitable for Ti6Al4V modified alloys for room temperature applications, it may not be suitable for creep applications to temperatures of 250 or 300C at which Ti6Al4V may be used, because Fe additions may lower creep resistance

Reviewer #3 (Remarks to the Author):

A new approach has been identified in this work to eliminate microstructural heterogeneity in Ti-6Al-4V, resulting from variations in thermal history during fabrication by laser powder bed fusion additive manufacturing, by modifying the alloy with cp-Ti and Fe₂O₃. The approach successfully eliminates heterogeneity and at the same time improves strength and ductility. The manuscript is well-written, but a few comments should be addressed before publication, as listed below:

- 1) It is explained that Fe addition favors the formation of beta phase owing to its beta stabilizing effect and higher diffusivity as compared to V, which rationalizes the addition of Fe₂O₃. However, the mechanism by which dilution of V through the addition of cp-Ti promotes beta phase formation is not clear. A follow-up question is, can addition of only Fe₂O₃, without any cp-Ti, eliminate the heterogeneity or not? This should be shown experimentally by printing Ti-6Al-4V + Fe₂O₃ alloy and performing the same microstructural characterization as done for other alloys. This result is also needed to support the authors' argument of synergistic effect of cp-Ti and Fe₂O₃ in eliminating the microstructural heterogeneity.
- 2) Provide the diffusivity values of V and Fe in alpha/martensite and beta phases at temperatures of interest.
- 3) A dedicated discussion on the sequence of phase transformation with and without the additives (cp-Ti and Fe₂O₃), possibly supported by a schematic, will give better insights into the mechanisms. It will help clarify questions such as: does the beta phase forms by martensitic decomposition or is it the retained beta from solidification; if beta forms due to martensitic decomposition, what is the contribution of accumulated heat in the sample, will the outcome change if the sample temperature of 500 °C assumed in the thermodynamic model is actually lower?
- 4) Although the SEM micrographs are able to differentiate between martensite and alpha + beta microstructures as a function of build height, these results can be supported by additional characterization using XRD, TEM, or both.
- 5) The best resolution for x-ray CT was 2 μm as mentioned, but SEM micrographs in supplementary Fig. 10 show many smaller pores. These smaller pores should be characterized as a function of the build height to strengthen the argument that the variation in ductility with change in build height is not due to porosity.
- 6) For every figure with tensile curves, fractographs, or x-ray CT data, please mention the location in the as-build part from where the characterized sample are extracted.
- 7) What liquid is used to suspend Fe₂O₃ particles?
- 8) Explain the meander scanning strategy in more detail.
- 9) Corrosion is included in the methods section and supplementary information, but not referred to in the main text.
- 10) There are many typos in the manuscript, please proofread carefully.

Response to Referees Letter

Designing against phase and property heterogeneities in additively manufactured titanium alloys

We would like to thank the editor for giving us the opportunity to revise the manuscript and thank all reviewers for their careful review and valuable comments. We have revised our manuscript accordingly and have highlighted all revisions in red in the revised manuscript. We have also added Prof. Gang Sha as one of the corresponding authors, due to his very important contribution to this work and the APT characterization. Please find below our point-by-point response to reviewer's comments, in which the reviewers' comments are in black and our responses are in blue. Additionally, the red texts indicate those added to the revised manuscript.

Point-by-point response to reviewer's comments

Reviewer #1:

In this manuscript, authors proposed a novel material designing strategy for additive manufacturing to achieve the phase and property uniformity in the AMed titanium alloy, using Ti-6Al-4V (Ti64) as an example. The critical and novel point in this strategy is to in-situ alloying of Ti64 with combined addition of pure titanium powders and iron oxide nanoparticle to tune the phase transformation kinetics during the additive manufacturing process. This strategy focusing on the uniformity of microstructure in the AMed titanium alloys is novel. In the past, the designing strategy of AMed titanium alloys mainly focused on the beta grain refinement (alloy design) or defect control (processing optimization). The current work on the other hand is focusing on the homogeneity of alpha+ beta microstructure in the AMed Ti-64 alloy. It is to avoid the alpha prime martensite phase formation during the

AM process in the layers close to the top of the product, where limited thermal cycles occurred. Thus, the designing strategy is novel and the reported research is significantly beneficial to the understanding of processing-microstructure-property in the AMed Ti64 alloy.

However, there are several questions authors are required to answer clearly:

Response: We really appreciate the reviewer for the careful review and positive comments.

1. The newly developed alloy is not Ti-64, but a new Ti-Al-V-Fe-O alloy. So why it is an important alloy to study?

Ti-64 alloy occupies the largest amount in the titanium market, and thus, significant amount of effort is spent on the low-cost manufacturing of this important Ti-64 alloy. However, in this work, different solutes (some is large amount) have been added into Ti-64 and thus significantly change the alloy composition. In the reported work, 25, 50 and 75wt% of CP-Ti was added into Ti-64 with 0.25-0.50wt% Fe_2O_3 . Thus, the alloy manufactured is not Ti-6-4 anymore, but a new Ti-Al-V-Fe-O alloy. So the question is even if the microstructure produced in the newly developed alloy is full of alpha + beta microstructure, why it is an important alloy or why people need to study and manufacture this alloy?

Response: We thank the reviewer for the in-depth discussion. We agree with the reviewer that titanium alloys produced by L-PBF in this work are not Ti-6Al-4V anymore. As the “workhorse” alloy in the titanium industry, Ti-6Al-4V was designed and optimized for the conventional manufacturing routes. Despite being well fabricated by AM with a very high density, Ti-6Al-4V does not mechanically perform to the best of its capacities. For example,

as shown in Fig. 3c, Ti-6Al-4V produced by L-PBF is generally less ductile than its solution-treated and aged (STA) counterpart. Besides, Ti-6Al-4V fabricated by DED and EB-PBF shows inferior strength to that under conventional STA condition. More importantly, Ti-6Al-4V produced by different AM technologies is often characterized by spatially dependent phases along the building direction, due to the unique thermal cycling of AM processes. The phase and associated property heterogeneities in additively manufactured Ti-6Al-4V may present great risks for applications under multiaxial stress states and variable amplitude loading conditions.

In this work, based on the understanding of how the elements contribute to the spatial phase distribution in Ti-6Al-4V, Ti-Al-V-Fe-O alloys were developed through rational alloy design. The newly developed titanium alloys are more compatible with the L-PBF process and exhibit uniform and superior mechanical properties compared with Ti-6Al-4V. Therefore, the driving force behind this work is to design new alloys specifically for the AM process with exceptional mechanical performance. The newly developed titanium alloys can be candidate materials for applications where titanium alloys with uniform mechanical properties are required.

To address this comment, we have revised the **Conclusion** (the last paragraph) of the main text on Page 11 to highlight: (1) the potential applications of the newly developed alloys, and (2), to a broader perspective, the influence of the alloy design strategy, as shown below.

“...We expect that the newly developed titanium alloys could be candidate materials for applications where titanium alloys with uniform mechanical properties are required. This requires a comprehensive evaluation of other mechanical properties (such as fatigue

properties and creep resistance) and corrosion resistance (Supplementary Note 2). Furthermore, unlike previous studies which have mainly focused on grain refinement (through alloy design) and/or defect control (via processing optimization), our work demonstrates that addressing the phase heterogeneity is of equal, if not greater, importance to achieve the desired uniform mechanical properties. Since the phase heterogeneity due to the solid-state thermal cycling has been reported in a wide variety of metallic materials fabricated by different AM technologies^{12,46-49}, we believe that our design strategy may help the development of other metallic alloys specifically for AM with uniform mechanical properties.”

2. The characterization of alpha' and alpha microstructure in the Ti-6-4 and newly designed Ti-Al-V-Fe-O alloy needs further analysis.

Alpha' and alpha phases exhibit HCP structure with slightly different composition (and thus different lattice parameter). It seems in the manuscript, morphology difference characterized in the SEM BSE imaging is mainly used to distinguish the alpha' and alpha phases (in Fig. 1, supplemental Fig. 1, supplemental Fig. 10 and supplemental Fig. 12). So did authors use APT study the composition of the claimed alpha prime phase in the Ti-64? If so, did authors observe any composition change in the alpha prime phase in the AMed Ti-64? If there is any composition change, can it be claimed as alpha prime phase? The reasons behind these questions actually challenge the proposed designing strategy, there may be limited diffusion occurs between the claimed alpha prime phase and beta matrix. If so, the addition of Fe, fast diffuser, is not that critical in forming alpha phase, as claimed by the authors.

Response: We thank the reviewer for the in-depth discussion. In this work, we utilized a FEI Scios Dual Beam SEM equipped with a concentric backscattered detector (CBS) to

differentiate between α' and α under a very low accelerating voltage (3 kV) and a small working distance (5.5 mm). Unlike conventional backscattered electrons (BSE) imaging, the contrast in different phases through this technology stems partially from electron channelling [1]. It is capable of detecting twins within α' (which are the fingerprint of α' martensite [2]) and has been recently used by researchers to distinguish α' and α [1, 3–5]. The main advantage of this technology over TEM is that the images can be obtained on bulk samples rather than thin foils. In addition, such SEM imaging allows for characterization of relatively large area on the sample. This is very important for this work to examine whether the microstructure is indeed homogeneous or not throughout the samples.

We didn't perform APT characterization of Ti-6Al-4V, because the previous work by Haubrich et al. [1] has provided a detailed systematic APT analysis of Ti-6Al-4V produced by L-PBF. It should be noted that Haubrich et al. carried out L-PBF on a SLM Solutions 280 HL machine and used Ti-6Al-4V powders with a chemical composition (**0.11 wt % O**, 6.4 wt % Al, 4.0 wt % V, 0.20 wt % Fe and balance Ti). Therefore, the APT work on Ti-6Al-4V by Haubrich et al. has provided very critical information to our study which used the AM machine (SLM Solutions 125 HL) from the same manufacturer and Ti-6Al-4V powders with a very close chemical composition (**0.08 wt % O**, 6.48 wt % Al, 4.06 wt % V, 0.21 wt % Fe, **0.01 wt % N**, **0.02 wt % C**, **0.003 wt % H** and balance Ti).

In their work, Haubrich et al. [1] found that martensite decomposition in Ti-6Al-4V – which is promoted by either intrinsic thermal cycling or post-AM heat treatment – involves substantial element partitioning. Specifically, *Al and O are accumulated in the α'/α phase, while V and Fe are rejected and diffuse into the β phase.* Accordingly, the accumulation of Al and O (which are α stabilizers) in the α' phase results in the formation of α phase, while the

addition of Fe (which is a fast diffuser and a β stabilizer) is important to form the β phase in a short period of time. This is consistent with other studies by Xu et al. [3,4] and Tan et al. [6] on additively manufactured Ti–6Al–4V by L-PBF and EB-PBF, respectively.

The kinetics of element partitioning is also captured by the DICTRA simulation in our work. We have plotted the composition profiles of Al and O across the α'/β interface at various times. It is evident that Al and O are accumulated within α' phase close to the α'/β interface. In the meantime, Fe and V diffuse out from α' phase to β phase. This dynamic element partitioning is in line with the conclusion drawn from the APT characterization by Haubrich et al. [1].

To address the concerns raised, we have also provided the DICTRA simulation of composition profiles of Al and O in Supplementary Fig. 14 of **Supplementary information** on Page 17.

Accordingly, we have also provided a short discussion on Supplementary Fig. 14 in **the revised manuscript** on Page 9.

“...On the other hand, it is found that the α stabilizers Al and O are accumulating in the α' phase as the cooling process proceeds (Supplementary Fig. 14a,b). The dynamic element partitioning of Fe, V, Al and O is in line with the conclusion drawn from the APT characterization by Haubrich et al. [27].”

Supplementary Fig. 14 | DICTRA simulation of the composition profiles of Al and O. a, composition profiles of Al and O across the α'/β interface at various times. **b,** Magnified view of the α' and β phase simulation domains (marked with black dashed rectangle in **a**) showing the accumulation of Al and O in the α' phase.

References

- [1] Haubrich, J. et al. The role of lattice defects, element partitioning and intrinsic heat effects on the microstructure in selective laser melted Ti-6Al-4V. *Acta Mater.* **167**, 136–148 (2019).
- [2] Pantawane, M.V. et al. Coarsening of martensite with multiple generations of twins in laser additively manufactured Ti6Al4V. *Acta Mater.* **213**, 116954 (2021).

- [3] Xu, W. et al. Additive manufacturing of strong and ductile Ti–6Al–4V by selective laser melting via in situ martensite decomposition. *Acta Mater.* **85**, 74–84 (2015).
- [4] Xu, W., Lui, E. W., Pateras, A., Qian, M. & Brandt, M. In situ tailoring microstructure in additively manufactured Ti–6Al–4V for superior mechanical performance. *Acta Mater.* **125**, 390–400 (2017).
- [5] Barriobero-Vila, P. et al. Mapping the geometry of Ti-6Al-4V: From martensite decomposition to localized spheroidization during selective laser melting. *Scr. Mater.* **182**, 48–52 (2020).
- [6] Tan X. P. et al. Graded microstructure and mechanical properties of additive manufactured Ti–6Al–4V via electron beam melting. *Acta Mater.* **97**, 1–16 (2015).

3. The claimed application of the approach to the beta titanium alloy needs further explanation.

Authors claim the proposed approach can be applied in the beta titanium alloys to trigger the omega phase and alpha phases in the building direction. However, Fe is a strong beta phase stabilizer and oxygen will impede the omega phase formation as well. Thus, I don't think the addition of Fe₂O₃ will promote the omega phase and alpha phase during the AM process in the building direction. So authors are required to introduce more details how the proposed strategy can be used in other titanium alloys.

Response: We thank the reviewer for this valuable comment. In fact, we took metastable β titanium alloy as an example to highlight “*The phase heterogeneity due to the solid-state thermal cycling has been reported in a wide variety of metallic materials fabricated by different AM technologies.*” Currently, we have been working on additively manufactured

metastable β titanium alloys and found that solid-state thermal cycling can also produce spatially dependent microstructures and hence undesirable non-uniform mechanical properties. We agree that the addition of Fe_2O_3 cannot address the phase heterogeneity in metastable β titanium alloys.

In this paragraph, we stated that “*we believe that our approach is not restricted to the selected titanium alloy presented here and could be applied to other metallic alloys*” in the initially submitted manuscript. It means that our alloy design strategy that aims at eliminating phase heterogeneity may help the design of other alloys which suffer from the same issue. The previous studies on achieving uniform mechanical properties have mainly focused on grain refinement and/or defect control. In fact, addressing the phase heterogeneity is of equal, if not greater, importance to grain refinement or defect control to eliminate the property heterogeneity.

To address the concerns raised, we have revised the **Conclusion** (the last paragraph) of the main text. Please see our response to Comment 1.

4. The addition of Fe, the claimed key to the proposed approach, needs further discussion.

Recently, different phase transformation mechanisms have been proposed in the field of titanium alloys to explain the formation of alpha phase. Whether or not partitioning is required to form alpha phase is being challenged:

1) Physical Review B 74, 134114 (2006). The concept of “bainitic alpha” was proposed in this work and it was claimed that “the growth of bainitic alpha plates is partitionless”.

2) Acta Materialia 60 (2012) 6247-6256. The concept of “pseudo-spinodal decomposition” was proposed that the structure and composition change in the formation of alpha may not occur simultaneously.

If diffusion is not required to form alpha microstructure in the titanium alloys, is it still necessary to add the fast diffuser (like Fe in the current work) or to manipulate the partitioning of alloying element in phase decomposition, which is the “key to our approach” claimed in the manuscript?

Response: We thank the reviewer for the helpful comment and the interesting references. We have read the above references carefully and still believe that elemental partitioning is critical to promote the formation of lamellar ($\alpha+\beta$) microstructure in our study. As mentioned in our response to **Comment 2**, the addition of Fe – which is a fast diffuser and a β stabilizer – is essential to promote the formation of β phase and hence the resulting lamellar ($\alpha+\beta$) microstructures. This is supported by previous studies on AM of Ti–6Al–4V [1–4]. For example, in the APT work by Haubrich et al. [1], it is found that increasing the volume energy density from 77 to 145 J/mm³ results in stronger intrinsic heat treatment effect, which leads to significant elemental partitioning of Fe, V, Al and O. Additional heat treatment of the part fabricated at 145 J/mm³ results in further elemental partitioning. Previous work on Ti–6Al–4V inspires us to find a pathway to address the phase heterogeneity through alloy design.

Table 4

Average element concentrations [at.%] determined from APT proximity histograms (HT = heat treated). For comparison: the Ti-6Al-4V feedstock consists of 10.8 at.% Al, 3.6 at.% V, 0.16 at.% Fe and 0.31 at.% O.

Phase	Elements	77 J/mm ³ , 'as- built' (Fig. S3)	145 J/mm ³ , 'as- built' (Fig. S4)	145 J/mm ³ , HT 400 °C (Fig. S5)	145 J/mm ³ , HT 530 °C (Fig. 9 and Fig. S6)
Regions enriched with β -stabilizers	V	10.0	16.0	20.0	18.0
	Fe	2.5	3.3	5.0	7.7
	Al	8.5	5.5	4.5	3.0
	O	0.4	0.3	0.1	0.2
α/α'	V	3.6	3.0	2.8	2.6
	Fe	0.2	0.2	0.1	0.1
	Al	10.2	9.5	9.8	9.6
	O	0.5	0.9	0.4	0.7
average concentration in the sample	V	3.6	3.5	3.8	3.5
	Fe	0.1	0.2	0.2	0.4
	Al	10.3	9.5	9.4	9.4
	O	0.3	0.4	0.4	0.7

Response Fig. 1 | The captured figure showing the measured element concentration of V, Fe, Al and O by APT in Haubrich et al.'s work. [1]

References

- [1] Haubrich, J. et al. The role of lattice defects, element partitioning and intrinsic heat effects on the microstructure in selective laser melted Ti-6Al-4V. *Acta Mater.* **167**, 136–148 (2019).
- [2] Xu, W. et al. Additive manufacturing of strong and ductile Ti-6Al-4V by selective laser melting via in situ martensite decomposition. *Acta Mater.* **85**, 74–84 (2015).
- [3] Xu, W., Lui, E. W., Pateras, A., Qian, M. & Brandt, M. In situ tailoring microstructure in additively manufactured Ti-6Al-4V for superior mechanical performance. *Acta Mater.* **125**, 390–400 (2017).
- [4] Tan X. P. et al. Graded microstructure and mechanical properties of additive manufactured Ti-6Al-4V via electron beam melting. *Acta Mater.* **97**, 1–16 (2015).

Reviewer #2 (Remarks to the Author):

This is an excellent contribution. The approach is novel, the methods and analysis is very well documented, the results on mechanical behavior quite interesting. My only recommendation for a minor modification is that the authors should point out also that while this approach is suitable for Ti6Al4V modified alloys for room temperature applications, it may not be suitable for creep applications to temperatures of 250 or 300C at which Ti6Al4V may be used, because Fe additions may lower creep resistance.

Response: We really appreciate the reviewer for the positive comments and constructive suggestions. The creep resistance of titanium alloys depends strongly on the volume fractions of α and β phases. In general, α phase shows superior creep resistance to β phase, due to the relatively limited ability for atoms to diffuse and HCP crystal structure [1]. The creep resistance of titanium alloys often deteriorates with increasing volume fraction of β phase. In this work, titanium alloys were developed based on Ti-6Al-4V through combined additions of CP-Ti and Fe₂O₃. As can be found in Supplementary Fig. 20 of the revised manuscript, the addition of CP-Ti to Ti-6Al-4V dilutes the concentration of V (the β stabilizer) and hence decreases the volume fraction of β phase, which is beneficial to creep resistance. On the other hand, the addition of Fe₂O₃ introduces both O and Fe. The former stabilizes α phase while the latter stabilizes β phase. We agree that Fe addition may deteriorate the creep resistance. However, it remains unclear whether the newly developed titanium alloys show lower or higher creep resistance than Ti-6Al-4V, because a combination of CP-Ti, O and Fe are introduced to Ti-6Al-4V. This will be explored in more details in the future work.

In the **Conclusion** (the last paragraph of the main text) of **the revised manuscript**, we have added a short discussion on the future work on Page 11, as shown below.

“...We expect that the newly developed titanium alloys could be candidate materials for applications where titanium alloys with uniform mechanical properties are demanded. This requires a comprehensive evaluation of other mechanical properties (such as fatigue properties and creep resistance) and corrosion resistance (Supplementary Note 2).”

References

- [1] Leyens, C. & Peters, M. (eds) Titanium and Titanium Alloys: Fundamentals and Applications (Wiley - VCH Verlag, 2003).

Reviewer #3 (Remarks to the Author):

A new approach has been identified in this work to eliminate microstructural heterogeneity in Ti-6Al-4V, resulting from variations in thermal history during fabrication by laser powder bed fusion additive manufacturing, by modifying the alloy with cp-Ti and Fe₂O₃. The approach successfully eliminates heterogeneity and at the same time improves strength and ductility. The manuscript is well-written, but a few comments should be addressed before publication, as listed below:

Response: We greatly thank the reviewer for the careful review and positive comments.

1) It is explained that Fe addition favors the formation of beta phase owing to its beta stabilizing effect and higher diffusivity as compared to V, which rationalizes the addition of Fe₂O₃. However, the mechanism by which dilution of V through the addition of cp-Ti promotes beta phase formation is not clear. A follow-up question is, can addition of only Fe₂O₃, without any cp-Ti, eliminate the heterogeneity or not? This should be shown experimentally by printing Ti-6Al-4V + Fe₂O₃ alloy and performing the same microstructural characterization as done for other alloys. This result is also needed to support the authors' argument of synergistic effect of cp-Ti and Fe₂O₃ in eliminating the microstructural heterogeneity.

Response: We thank the reviewer for pointing out this important omission. We fabricated (Ti-6Al-4V + Fe₂O₃) alloys according to the research plan. However, we did not include the associated result in the initially submitted manuscript, because phase homogeneity cannot be achieved through the sole addition of Fe₂O₃ and we thought that it was not directly related to our alloy design strategy (that is, the combined addition approach). This comment makes us

realize that it is very important to include this result to demonstrate the synergistic effect of CP-Ti and Fe₂O₃.

To address this comment, we have provided the microstructural characterization of Ti-6Al-4V + 0.25 wt % Fe₂O₃ in Supplementary Fig. 21 of **Supplementary information**.

Supplementary Fig. 21 | Microstructures of (Ti-6Al-4V + 0.25 wt % Fe₂O₃) along the building direction. a, SEM-BSE images showing the microstructures in the fabricated part at different locations along the building direction (BD). **b,** Higher magnification of the selected regions in **a.** **c,** Higher magnification of the selected regions in **b.**

Accordingly, we have provided a short discussion on Supplementary Fig. 21 in Supplementary Note 5 of **Supplementary information** on Page 31–32.

“Supplementary Fig. 21 shows the microstructural analysis of (Ti–6Al–4V + 0.25 wt % Fe₂O₃) along the building direction. Overall, the addition of 0.25 wt % Fe₂O₃ to Ti–6Al–4V results in a significant reduction in the phase width throughout the part compared with Ti–6Al–4V (Supplementary Fig. 1a,b). It is apparent that the internally twinned martensite remains in the top region (Position A and Position B) and a fully lamellar ($\alpha+\beta$) microstructure forms in the lower region (Position D), indicating that the sole addition of Fe₂O₃ to Ti–6Al–4V cannot eliminate the phase heterogeneity.

In summary, by comparing the microstructures of (Ti–6Al–4V + 50 wt % CP–Ti) (Supplementary Figs. 20a,b) and (Ti–6Al–4V + 0.25 wt % Fe₂O₃) to those of 50Ti–0.25O (Supplementary Fig. 11c,d), it suggests that CP–Ti and Fe₂O₃ have a synergy in their contributions to the homogeneous lamellar ($\alpha+\beta$) microstructures.”

2) Provide the diffusivity values of V and Fe in alpha/martensite and beta phases at temperatures of interest.

Response: We thank the reviewer for the valuable suggestion. We have provided the temperature dependence of diffusivities of V and Fe in β and α phases in Supplementary Fig. 13. This figure was mentioned in the main text of **the revised manuscript** on Page 9, as shown below.

“It is evident that Fe shows a much stronger partitioning tendency in the β phase than V (Supplementary Fig. 12 c-e), due to its significantly high diffusivity (Supplementary Fig. 13).”

Supplementary Fig. 13 | Temperature dependence of diffusivities of Fe and V in β and α phases. The diffusivity values of Fe and V were calculated using Thermo-Calc software implemented TCTI3 and MOBTI4 database.

3) A dedicated discussion on the sequence of phase transformation with and without the additives (cp-Ti and Fe_2O_3), possibly supported by a schematic, will give better insights into the mechanisms. It will help clarify questions such as: does the beta phase forms by martensitic decomposition or is it the retained beta from solidification; if beta forms due to martensitic decomposition, what is the contribution of accumulated heat in the sample, will the outcome change if the sample temperature of 500 °C assumed in the thermodynamic model is actually lower?

Response: This is a great suggestion! We have provided a schematical diagram showing the phase transformation pathways in Ti-6Al-4V and the newly developed alloys. The associated discussion is provided in **Supplementary Note 6**.

Supplementary Fig. 22 | Schematic illustration of phase transformation pathways for L-PBF produced Ti-6Al-4V and the newly developed alloys under the thermal cycling. **a**, The L-PBF process. **b**, The thermal cycling of a location (marked with orange box J) experienced during L-PBF due to the track-by-track and layer-by-layer fabrication. **c**, The sequence of phase transformation at different stages (A, B, C and D marked in **b**) in Ti-6Al-4V and the newly developed alloys.

4) Although the SEM micrographs are able to differentiate between martensite and alpha + beta microstructures as a function of build height, these results can be supported by additional characterization using XRD, TEM, or both.

Response: We thank the reviewer for the valuable suggestions. We have provided XRD and TEM characterization to support the phase analysis by SEM. As shown in Supplementary Fig. 2, the XRD result of Ti-6Al-4V samples shows the phase heterogeneity along the building direction and is consistent with SEM analysis (Fig. 1b, Supplementary Figs. 1a and 1b). The

XRD result is mentioned in the main text of **the revised manuscript** on Page 3, as shown below.

“...Such a graded phase distribution is also confirmed by scanning electron microscope (SEM) (Fig. 1b, Supplementary Figs. 1a and 1b) and X-ray Diffraction (XRD) (Supplementary Fig. 2) in this work.”

Supplementary Fig. 1 | X-ray diffraction spectra of L-PBF produced Ti-6Al-4V samples taken from different locations along the building direction of the fabricated part. It is evident that the specimen on the top surface (Specimen H1) exhibits a single HCP phase while specimens in the lower regions (Specimens H2, H4 and H6) gradually show the presence of BCC phase.

We have also added the experimental procedure on XRD in **Methods** of **the revised manuscript** on Page 24, as shown below.

“X-ray diffraction

X-ray diffraction (XRD) analysis was conducted on a D8 ADVANCE X-ray diffractometer (Bruker, Germany) with Cu radiation source operated at 40 kV and 40 mA with a step size of 0.02°.”

Besides, we have provided the TEM images of the selected newly developed alloy, as shown in Supplementary Fig. 15. This figure is mentioned in the main text of **the revised manuscript** on Page 9, as shown below.

“We observed the lamellar ($\alpha+\beta$) microstructure in the selected newly developed alloy (50Ti–0.50O) using transmission electron microscopy (TEM) (Supplementary Fig. 15a,b). The STEM-EDS images clearly reveal that the β phase is enriched in both Fe and V while it is depleted in Al (Supplementary Fig. 15c).”

The experimental information on TEM sample preparation and TEM characterization is also provided in **Methods of the revised manuscript** on Page 25.

“Transmission electron microscopy

Samples for transmission electron microscope (TEM) observations were prepared using a FEI Scios Dual Beam system (Thermo Fisher Scientific Inc., USA). TEM characterization was performed on a FEI Tecnai G2 F20 TEM (Thermo Fisher Scientific Inc., USA) operated at an acceleration voltage of 200 kV in both TEM and scanning TEM (STEM) modes.”

Supplementary Fig. 15 | TEM observation of the L-PBF fabricated 50Ti–0.50 alloy. a, Bright-field TEM image showing the presence of fine β plates between coarse α laths. **b,** Dark-field image corresponding to **a** showing the β phase. **c,** HADDF-STEM image and EDS elementary mapping (Fe, V, Al and O).

5) The best resolution for x-ray CT was 2 μm as mentioned, but SEM micrographs in supplementary Fig. 10 show many smaller pores. These smaller pores should be characterized as a function of the build height to strengthen the argument that the variation in ductility with change in build height is not due to porosity.

Response: We thank the reviewer for pointing out this important issue. The small “pores” in Supplementary Fig. 10 of the initially submitted manuscript are in fact the silica (SiO_2) nanoparticles (with the particles size of 0.25 μm) from Struers OP-S NonDry suspension in the sample preparation process, instead of real “pores” that were introduced during

fabrication. Please find below the backscattered electrons (BSE) and the corresponding secondary electrons (SE) images of one sample. It is shown that the residual silica nanoparticles result in the appearance of pores in the BSE micrograph.

Response Fig. 2 | **a**, SEM backscattered electrons (BSE) and **b**, the corresponding secondary electrons (SE) images showing that SiO₂ nanoparticles result in the appearance of pores.

To address this comment, we have reperfomed the SEM-BSE imaging on the sample and have replaced the misleading SEM images with the new ones, as shown in Supplementary Fig. 11 in **Supplementary information**.

Supplementary Fig. 11 | Microstructures of 75Ti–0.25O and 50Ti–0.25O alloys at different magnifications. a, SEM-BSE micrographs of different locations in the 75Ti–0.25O part along the building direction (BD). **b,** Higher magnification of the selected regions in **a**. The height of the L-PBF produced part is 40 mm. The distance between two characterization locations is about 7 mm. **c,** SEM-BSE images of different locations in the 50Ti–0.25O alloy part. **d,** Higher magnification of the selected regions in **c**.

6) For every figure with tensile curves, fractographs, or x-ray CT data, please mention the location in the as-built part from where the characterized sample are extracted.

Response: We thank the reviewer for pointing out this very important omission. We have added this information in the figures and figure captions of Ti–6Al–4V which shows significant difference in the microstructure and mechanical property along the building direction. In the case of the newly developed alloys, the phase and property heterogeneities have been addressed. Therefore, we do not add this information in the figures of tensile curves (like what we did in Fig. 1c, that is, H1, H2, H3...), because this will make the figures look somewhat messy. However, we have provided this information in the text and other figures.

7) What liquid is used to suspend Fe₂O₃ particles?

Response: We thank the reviewer for this important omission. The liquid to suspend Fe₂O₃ particles is deionized water. We have provided this information in **Feedstock preparation in Methods of the revised manuscript** on Page 22.

8) Explain the meander scanning strategy in more detail.

Response: We thank the reviewer for this comment. Please find below a schematical illustration of the meander scanning strategy. We have provided this figure in **Supplementary information** as Supplementary Fig. 17.

Supplementary Fig. 17 | The meander scanning strategy used in the present work.

9) Corrosion is included in the methods section and supplementary information, but not referred to in the main text.

Response: We thank the reviewer for pointing out this important omission. We have included the corrosion information in the last paragraph of the main text on Page 11, as shown below.

“...We expect that the newly developed titanium alloys could be candidate materials for applications where titanium alloys with uniform mechanical properties are demanded. This

also requires a comprehensive evaluation of other mechanical properties (such as fatigue properties and creep resistance) and corrosion resistance (Supplementary Note 2).”

10) There are many typos in the manuscript, please proofread carefully.

Response: We thank the reviewer for pointing out the language issues. We have tried our best to carefully proofread the revised manuscript and supplementary information.

REVIEWERS' COMMENTS

Reviewer #1 (Remarks to the Author):

Authors have made up all the corrections as suggested. The manuscript is ready to be published.

Reviewer #3 (Remarks to the Author):

The authors have addressed all the comments satisfactorily. The manuscript is recommended for publication.